# On the added value of sequential deep learning for upscaling of evapotranspiration

Basil Kraft[1,2,3], Jacob A. Nelson[1], Sophia Walther[1], Fabian Gans[1], Ulrich Weber[1], Gregory Duveiller[1], Markus Reichstein[1], Weijie Zhang[1], Marc Rußwurm[5], Devis Tuia[4], Marco Körner[3], Zayd Hamdi[1], and Martin Jung[1]

[1]Max Planck Institute for Biogeochemistry (MPI BGC)
[2]Swiss Federal Institute of Technology Zurich (ETH)
[3]Technical University of Munich (TUM)
[4]École Polytechnique Fédérale de Lausanne (EPFL)
[5]Wageningen University (WU)

**Correspondence:** Basil Kraft (basil.kraft@env.ethz.ch)

**Abstract.**

Estimating ecosystem-atmosphere fluxes such as evapotranspiration (ET) in a robust manner and at global scale remains a challenge. Machine learning–based methods have shown promising results to achieve such upscaling, providing a complementary methodology that is independent from process-based and semi-empirical approaches. However, a systematic evaluation of the skill and robustness of different machine learning (ML) approaches is an active field of research that requires more investigations. Concretely, deep learning approaches in the time domain have not been explored systematically for this task.

In this study, we compared instantaneous (i.e., non-sequential) models—extreme gradient boosting (XGBoost) and a fully-connected neural network (FCN)—with sequential models—a long short-term memory (LSTM) model and a temporal convolutional network (TCN)—for the modeling and upscaling of ET. We compared different types of covariates (meteorological without precipitation, precipitation, remote sensing, and plant functional types) and their impact on model performance at the site level in a cross-validation setup.

When using only meteorological covariates, we found that the sequential models (LSTM and TCN) performed better—each with a Nash-Sutcliffe modeling efficiency (NSE) of 0.73—than the instantaneous models (FCN and XGBoost)—both with an NSE of 0.70—in site level cross-validation at the hourly scale. The advantage of the sequential models diminished with the inclusion of remote-sensing-based predictors (NSE of 0.75 to 0.76 versus 0.74). On the anomaly scale, the sequential models consistently outperformed the non-sequential models across covariate setups, with an NSE of 0.36 (LSTM) and 0.38 (TCN) versus 0.33 (FCN) and 0.32 (XGBoost) when using all covariates.

For the upscaling from site to global coverage, we input the two best-performing combinations of covariates—meteorological and remote sensing observations, and with precipitation and plant functional types in addition—with globally available gridded data. To evaluate and compare the robustness of the modeling approaches, we generated a cross-validation-based ensemble of upscaled ET, compared the ensemble mean and variance among models, and contrasted it with independent global ET data. In particular, we investigate three questions regarding the performance of sequential models compared to the non-sequential

models in the context of spatial upscaling: a) whether they lead to more realistic and robust global and regional ET; b) whether they are able to capture the temporal dynamics of ET better; and c) how robust they are to the covariate setup and training data subsets.

The generated patterns of global ET variability were relatively consistent across the ML models overall, but in regions with low data support via EC stations, we observed substantial biases across models and covariate setups, and large ensemble uncertainties. The sequential models better capture temporal dynamics of ET when upscaled to global coverage, especially when using precipitation as additional input, and they seem to be more robust to covariate setups, particularly the LSTM model. However, they exhibited, together with the non-temporal FCN model, larger ensemble spread than XGBoost, and they yielded lower global ET estimates than what is currently understood. XGBoost showed smaller ensemble spread compared to neural networks in particular when conditions were poorly represented in the training data, but it was more sensitive to the covariate setup. Plant functional types were useful at site level for improved representation of spatial patterns, but had a large leverage on upscaling results—i.e., having disproportionate impact on the spatial patterns—especially for XGBoost, but less for the LSTM model.

Our findings highlight non-linear model responses to biases in the training data and underscore the need for improved upscaling methodologies, which could be achieved by increasing the amount and quality of training data or by the extraction of more targeted features representing spatial variability. The neural networks seem to yield more realistic ensemble uncertainty compared to XGBoost. Approaches such as transfer learning, knowledge-guided ML, or hybrid modeling, which encourage physically consistent results while harnessing the efficiency of ML, should be further investigated. Deep learning for flux upscaling holds large promise, while remedies for its vulnerability to training data distribution changes still need consideration by the community.

## 1 Introduction

Measurements of land-atmosphere fluxes of gases, such as water vapor or carbon dioxide, are crucial for understanding the interactions between climate and ecosystems. Instruments at eddy covariance (EC) stations measure such fluxes integrated over a time span of 30 or 60 minutes and a small spatial footprint, spanning a couple of hundred meters to over a kilometer, depending on the instrument height, terrain roughness, and wind conditions. The measurement is performed at *ecosystem level*, as it represents the integral of biotic and abiotic processes across scales (Baldocchi et al., 2001). While EC stations provide a crucial source of data to measure these fluxes, they come with challenges. For instance, their representativeness and applicability for regional to global analysis may be restricted due to the sparsity of EC sites in geographic and climate space (Fig. 1a and b).

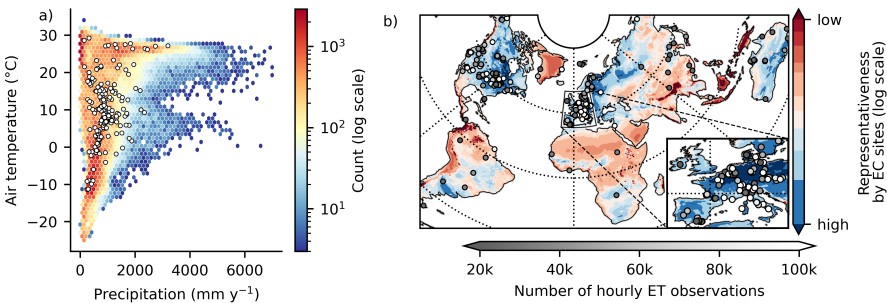

**Figure 1.** Overview of eddy covariance (EC) sites used in this work: **a)** Distribution of EC sites (white points) and map grid-cells (background color) within the global climate in terms of mean temperature and annual precipitation. **b)** Geographic EC site locations in different gray scales according to the number of hourly observations of evapotranspiration. The map color corresponds to the representativeness of a geographic location by the EC station sites. It is the average Euclidean distance in climate space (mean and standard deviation of normalized 15-daily temperature, precipitation, and radiation) to the ten closest stations. A lower representativeness (red) means a given location is further away from EC sites in climate space.

Evapotranspiration (ET) is the combined flux of water vapor via evaporation from bare surfaces and plant transpiration. The ET flux is of high relevance for modeling and understanding the Earth system because it links water, carbon, and energy cycles (Jung et al., 2010; Nelson et al., 2018). However, the modeling of ET is challenging due to the highly dynamic nature and modulation by ecosystems. Their behavior depends on past system exposure via so-called dynamic memory effects (Ogle et al., 2015; Besnard et al., 2019; Kraft et al., 2019, 2021). Among other factors, ET depends on soil moisture, which is primarily driven by the recent past rather than by instantaneous weather conditions. Other processes impacting ET that depend on past meteorology are related to vegetation states, such as the leaf area or phenology (Migliavacca et al., 2012).

To consider such complex memory effects, a model must incorporate past system exposure, such as temperature or precipitation. Alternatively, the model can be fed with states that represent past exposure, such as leaf area index (LAI) and soil moisture observations, or aggregations of past meteorology like temperature or precipitation sums. However, the observation of ecosystem states is challenging and often not possible. *In-situ* measurements, e.g., of soil moisture, are not consistently available at all EC stations and may not always precisely coincide with the eddy covariance measurements in space or time, limiting the applicability for across-site modeling. As an alternative, remotely sensed observations can serve as proxies of ecosystem states, like vegetation indices for foliage or phenology. These observations alone can only partially explain EC measurements, as they represent structural or optical properties of the canopy rather than plant physiology or subsurface water states, and especially optical observations tend to saturate with dense vegetation (Huete et al., 2002). Therefore, it may be beneficial to *learn* the non-observable states for the modeling of land-atmosphere fluxes as non-linear functions of available covariates. Here, sequential machine learning (ML) models may offer a unique opportunity, as they are able to extract dynamic proxies from temporal data (Rußwurm and Körner, 2017; Kraft et al., 2019).

ET can be quantified at large scales employing process-based paradigms, i.e., land surface models, or semi-empirical approaches, based on inputs from remote sensing observations and predefined empirical relationships (e.g., the Global Land Evaporation Amsterdam Model (GLEAM), Martens et al., 2017). As a complementary approach, the data-driven upscaling, i.e., the generalization from the irregularly distributed EC stations to a regular spatio-temporal field, can provide independent insights into ecosystem processes (Jung et al., 2017). The upscaling is achieved by training an ML model at the EC sites with covariates that are also available as spatio-temporal fields (Jung et al., 2009). The optimized model is then fed with the contiguous covariates to generate regional to global scale products.

Due to the availability of long-term records of both eddy covariance data and remote sensing products, increased computational capacities, and a higher acceptance of ML approaches in the geosciences (Camps-Valls et al., 2021), data-driven approaches to model ecosystem-atmosphere fluxes have gained momentum in the past decade (Tramontana et al., 2016; Jung et al., 2011; Nelson & Walther et al., 2024; Zhu et al., 2024). Today, ML is widely used to model and upscale EC data, but the field is still dominated by non-sequential modeling (i.e., an instantaneous model that does not learn memory effects), such as decision trees or fully-connected neural networks.

An ensemble of global, harmonized products of upscaled EC fluxes from different ML algorithms (tree, kernel, regression splines, and neural network-based methods) was released by the FLUXCOM initiative (FLUXCOM, 2017), founded on previous work by Beer et al. (2010), Jung et al. (2010, 2011), and Tramontana et al. (2016). These products are built upon non-sequential models, and they account for memory via manually designed features, such as seasonal amplitudes or water availability indices, and remote sensing-based ecosystem state proxies, like vegetation indices (Huete et al., 2002). The FLUXCOM products of energy (Jung et al., 2019) and carbon (Jung et al., 2020) are utilized in contemporary land-atmosphere interaction studies and function as benchmarks for Earth system models. To improve the temporal resolution and resolve the diurnal cycle, Bodesheim et al. (2018) upscaled 30-minute fluxes of carbon and energy using randomized decision forests (Breiman, 2001), with a non-sequential modeling approach. Xiao et al. (2014) upscaled daily carbon and water fluxes in North America using moderate imaging spectroradiometer (MODIS) data with non-sequential ML approaches. Xu et al. (2018) evaluated different non-sequential ML methods to upscale ET with high-resolution features available regionally in China. Zhao et al. (2019) and ElGhawi et al. (2023) both used a non-sequential physics-constrained neural networks approach to model ET, which has the potential to yield physically consistent and partially interpretable models. Recently, Nelson & Walther et al. (2024) published an hourly upscaling product of carbon and energy fluxes (X-BASE), built upon a novel framework (FLUXCOM-X), which enables the testing and application of different data streams and ML methods for upscaling in a flexible manner. They use a non-sequential model based on boosted regression trees (XGBoost; Chen and Guestrin, 2016) and account for memory effects via remote sensing state proxies.

Non-sequential ML approaches, however, cannot represent temporal variable interactions beyond the observable state proxies in contrast to, for instance, recurrent neural networks (RNNs; Lipton et al., 2015). For time series regression, the long short-term Memory network (LSTM; Hochreiter and Schmidhuber, 1997) is a widely used architecture based on the RNN paradigm (Van Houdt et al., 2020). Such sequential approaches have been evaluated for EC flux modeling at the site level. Reichstein et al. (2018) applied RNNs to model weekly net ecosystem exchange of carbon (NEE) from 9 European flux stations

with meteorological forcing and showed the relevance of temporal information via a permutation test. Besnard et al. (2019) employed an LSTM architecture to model monthly NEE at EC sites and achieve better performance than with a non-sequential random forest. But still, they reported poor representation of temporal dynamics both in terms of interannual variability and anomalies, the deviations from the mean seasonal cycle.

In the domain of deep learning, different model architectures are capable of processing sequential data. In the Earth sciences, the LSTM has become the *de facto* standard, even though other architectures have been developed, such as the temporal convolutional network (TCN; Oord et al., 2016; Bai et al., 2018). The TCNs use sparse convolution along the temporal dimension to consider long-term effects more efficiently. More recently, models employing self-attention (Vaswani et al., 2017) have shown noteworthy performance in many domains. These sequential models could also hold potential for EC flux modeling, as has been shown by Armstrong et al. (2022) and Nakagawa et al. (2023). While conceptually apparent, there is little systematic evidence of whether such sequential deep learning methods provide an advantage over non-sequential approaches for the upscaling of EC fluxes, and about how these models respond to other issues inherent in upscaling, such as limited and unevenly sampled training data and distribution shift from the local point data to gridded fields.

In this study, we provide a systematic comparison of different machine-learning approaches to the modeling of site-level ET fluxes and subsequently their upscaling to the global scale using the FLUXCOM-X modelling framework (Nelson & Walther et al., 2024). A simple linear model, XGBoost, and a feed-forward fully connected neural network serve as baselines for non-sequential models. Two sequential models, one based on the LSTM architecture, and another based on a TCN, account for temporal effects. We chose these models as they are conceptually different but both commonly used for time-series simulation and forecasting tasks, and we acknowledge that other architectures could be used as well. We compare the model performances at the site level in a cross-validation setup and assess the relevance of dynamical memory effects for ET modeling. For each model, we conduct a feature ablation experiment, where we drop feature groups. The groups considered in addition to meteorology are precipitation (which we obtained from reanalysis data and not from site-level observations due to large gaps), dynamic state representations, based on remotely sensed observations, and plant functional types (PFTs), which are static descriptors of site characteristics. We provide and investigate cross-validation–based upscaling ensembles from the independent cross-validation models to test for robustness. To assess the impact of the model architecture on upscaling, we contrast our products globally to a set of land surface model simulations and to a semi-empirical approach (GLEAM). We investigate, in the context of upscaling, a) whether the sequential models lead to more realistic and robust global and regional ET compared to independent estimates, b) whether they are able to capture the temporal dynamics of ET better, and c) how robust they are to the covariate setup and training data subsets, compared to the non-sequential models.

The key contributions of this study are:

- A systematic comparison of the effectiveness of different ML methods for site-level land-atmosphere ET flux modeling.

- An assessment and discussion of the relevance of different covariates in the context of ecological memory effects for ET.

- A characterization and comparison of an ensemble of upscaled ET estimates generated with different ML models.

## 2   Data sources and processing

We used hourly EC data from 2001 to 2020 processed by the ONEFLUX pipeline (Pastorello et al., 2020). Only sites available under the CC BY 4.0 license were included in this analysis, i.e., FLUXNET 2015 (Pastorello et al., 2020), ICOS Drought 2018 (Drought 2018 Team and ICOS Ecosystem Thematic Centre, 2020), ICOS Warm Winter 2020 (Warm Winter 2020 Team and ICOS Ecosystem Thematic Centre, 2022), or more recent ICOS or Ameriflux releases when present. In total, we used 287 sites with approximately 19 million hourly observations of ET and meteorological conditions. The approach by Jung et al. (2023) was used for quality flagging. We used latent heat energy as target flux and convert it to ET assuming a constant latent heat of vaporization of $2.45 \mathrm{\ MJ\,mm^{-1}}$. The following meteorological covariates were considered: near-surface air temperature ($T_\mathrm{air}$), vapor pressure deficit ($\Delta e$), shortwave irradiation ($R_\mathrm{in}$), potential shortwave irradiation ($R_\mathrm{in,\,pot}$), and time-derivative of potential shortwave irradiation ($\Delta R_\mathrm{in,\,pot}$). Note that the time-derivative, which is the difference between potential shortwave irradiation values for two consecutive hours, are intended to help the non-sequential models discern the diurnal cycle. As precipitation ($P$) observation at site level is often missing, we used the hourly ERA5 reanalysis instead (Hersbach et al., 2020), extracted from the nearest pixel to the site. In addition, we used remote sensing observations from the moderate imaging spectroradiometer (MODIS) sensor on board both Terra and Aqua satellite platforms, collection 006. These include the enhanced vegetation index (EVI,  Huete et al., 2002) and the normalized difference water index (NDWI,  Gao, 1996), both retrieved at site level from the MCD43A4 product and quality filtered based on the MCD43A2 product (spatial resolution of 500 m, Schaaf and Wang, 2015a, b), and from MCD43C4 for the global data runs (spatial resolution of 0.05 °, Schaaf and Wang, 2015c). Additionally, the land surface temperature (LST) was obtained from MOD11A1 at site level (Wan et al., 2015a, spatial resolution of 1km), and from MOD11C1 globally (Wan et al., 2015b, spatial resolution of 0.05°). Although the MCD43A4 product for the reflectances uses observations from a period of 16 days to characterize and invert the bidirectional reflectance distribution function of a given pixel for the day at the center of the period, this operation is done over a temporally moving window at daily timesteps, resulting in output data with daily frequency. Processing of the datasets, cutouts at the sites, and quality control correspond to the set-up used in the FLUXCOM-X-BASE data set (Nelson & Walther et al., 2024; Walther & Besnard et al., 2022; Jung et al., 2023). As an optional covariate, we use the plant functional type (PFT), available for all EC station sites. The nine PFTs were one-hot-encoded and repeated in time to match the hourly time series. One-hot encoding represents categorical variables as binary values, assigning a unique binary digit to each category. Sample time series of the covariates and ET are shown in Fig. 2.

For upscaling, we used global meteorological data from the ERA5 reanalysis (Hersbach et al., 2020) corresponding to the site level variables. The hourly data was spatially resampled to a resolution of 0.05° using bi-linear interpolation. This data was also used to fill gaps in site-level meteorological observations.

For the evaluation of the upscaling results, due to the lack of direct and spatially contiguous observations of ET, we used the Global Land Evaporation Amsterdam Model (GLEAM) v3 (Martens et al., 2017) and global sums of yearly ET from 14 land surface modes (TRENDY v6, values extracted from Pan et al., 2020) as reference. Note that these reference data sources do not

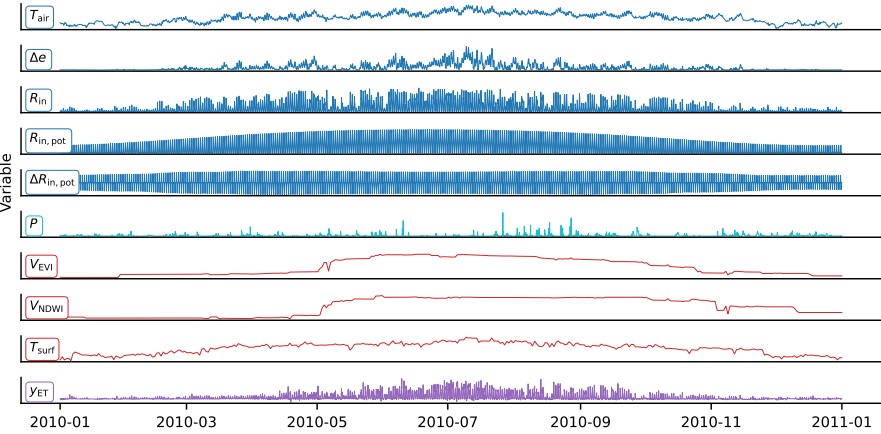

**Figure 2.** One-year time series from the Hainich site (DE-Hai) in Germany. Meteorological covariates (hourly): near-surface air temperature ($T_{\text{air}}$), vapor pressure deficit ($\Delta e$), shortwave irradiation ($R_{\text{in}}$), potential shortwave irradiation ($R_{\text{in, pot}}$), time-derivative of potential shortwave irradiation ($\Delta R_{\text{in, pot}}$), and precipitation ($P$). Remote sensing (daily): enhanced vegetation index ($V_{\text{EVI}}$), normalized difference water index ($V_{\text{NDWI}}$), and land surface temperature ($T_{\text{surf}}$). Land-atmosphere target flux (hourly): evapotranspiration ($y_{\text{ET}}$).

represent the ground truth, but are estimates derived using different approaches, independent from the data-driven upscaling performed here.

## 3  Methods

### 3.1  Experimental setup

We evaluate a set of sequential and non-sequential ML models at the site level in a leave-sites-out cross-validation setup. The models are trained with different types of covariates: meteorological site-level observations without precipitation (`met`), precipitation from ERA5 (`prec`), remote sensing (`rs`), and PFTs (`pft`). These experiments with different sets of variables as model inputs are summarized in Tab. 1. In total, six covariate setups were tested and combined with five machine-learning models, i.e., thirty models were trained and evaluated at the site level. For the evaluation, we use the Nash-Sutcliffe modeling efficiency (Nash and Sutcliffe, 1970)

$$\text{NSE} = 1 - \frac{\sum_{t=1}^{T}(y_t - \hat{y}_t)^2}{\sum_{t=1}^{T}(y_t - \bar{y})^2} \quad , \tag{1}$$

where $y_t$ is the observed and $\hat{y}_t$ the predicted ET at time $t$, and $\bar{y}$ represents the mean of the observations. The NSE is calculated per site and can take values from $-\infty$ to $1$ and reflects model performance relative to the mean of the observations. Values above $0$ indicate better prediction than using the mean observations, and $1$ is a perfect prediction. Note that for the evaluation of spatial patterns, the NSE was not computed per site but across sites, which corresponds to the $R^2$.

**Table 1.** The ablation experiment with different covariate groups: Meteorological without precipitation (`met`, hourly), precipitation (`prec`, hourly), remote sensing–based (`rs`, daily), and plant functional type (`pft`, constant). Each item corresponds to a unique covariate setup.

| Setup | Covariates |
|---|---|
| `met` | $T_{\text{air}}, \Delta e, R_{\text{in}}, R_{\text{in, pot}}, \Delta R_{\text{in, pot}}$ |
| `met+prec` | met $+ P$ |
| `met+pft` | met $+ S_{\text{PFT}}$ |
| `met+rs` | met $+ V_{\text{EVI}}, V_{\text{NDWI}}, T_{\text{surf}}$ |
| `met+pft+rs` | met $+ V_{\text{EVI}}, V_{\text{NDWI}}, T_{\text{surf}} + S_{\text{PFT}}$ |
| `met+prec+pft+rs` (`=full`) | met $+ P + V_{\text{EVI}}, V_{\text{NDWI}}, T_{\text{surf}} + S_{\text{PFT}}$ |

`met`: near surface air temperature $T_{\text{air}}$; vapor pressure deficit $\Delta e$; shortwave irradiation $R_{\text{in}}$; potential shortwave irradiation $R_{\text{in, pot}}$; time-derivative of the potential shortwave irradiation $\Delta R_{\text{in, pot}}$; `prec`: precipitation $P$; `rs`: enhanced vegetation index $V_{\text{EVI}}$; normalized difference water index $V_{\text{NDWI}}$; land surface temperature $T_{\text{surf}}$; `pft`: plant functional type $S_{\text{PFT}}$.

## 3.2 Modeling approach

With the goal of evaluating model performance at EC station locations and afterwards upscaling to the global scale, we tested a number of ML algorithms in a site-level cross-validation setup. We denote the modeling problem as

$$\hat{y}_{s,t} = f_\theta(\boldsymbol{X}_{s,t-K:t}, \boldsymbol{c}_s) \quad . \tag{2}$$

Here, $\boldsymbol{X}_{s,t-K:t} \in \mathbb{R}^{(K+1) \times D}$ are the $D$ dynamic input covariates with up to $K$ antecedent time steps, and $\boldsymbol{c}_s \in \mathbb{R}^M$ are the $M$ static (constant) input features. The target flux of ET is represented as $\hat{y}_{s,t} \in \mathbb{R}$ at site $s$ and time step $t$. Note that $K = 0$ with only instantaneous covariates $\boldsymbol{X}_{s,t}$ is a special case where no antecedent time steps are considered (i.e., a non-sequential model). We aim to find the parameters

$$\theta^* = \underset{\theta}{\arg\min} \, \mathcal{L}(f_\theta(\boldsymbol{X}_{s,t-K:t}, \boldsymbol{c}_s), \boldsymbol{y}_t) \tag{3}$$

of a function $f_\theta$ that minimize the loss function $\mathcal{L}$, given by the mean squared error (MSE).

As baselines, we used a linear regression (`linearreg`) as well as two non-sequential models, a fully connected feed-forward neural network (`fcn`), and extreme gradient boosting (`xgboost`). The latter was also used in the recent state-of-the-art global upscaling product `xbase` (Nelson & Walther et al., 2024). The setup for these models was largely consistent with `xbase`, i.e., the same covariates were used here plus precipitation. The remote sensing and PFT covariates were repeated in time for every hour to obtain uniform inputs, i.e., the former were constant over a day and the latter over the entire time series. Although the remote sensing covariates do, in theory, vary on a sub-daily basis, these variables are not available at the hourly resolution, and the diurnal variations are driven primarily by hourly-varying meteorological variables, though they interact with satellite-based features that change only on a daily to weekly basis. In addition to these non-sequential models, we used two

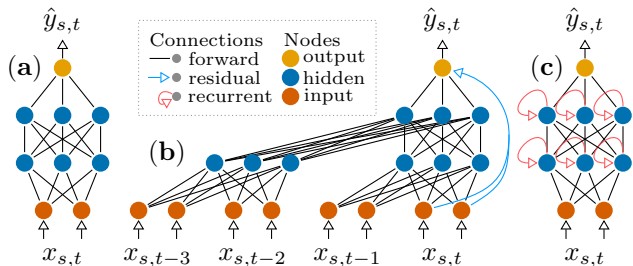

**Figure 3.** The three neural network layers used in this study: **a)** a feed-forward neural network, **b)** a temporal convolutional network (TCN), which applies causal (i.e., does not consider future time steps) 1D convolutions in the time dimension and **c)** a long short-term memory (LSTM) model, which uses recursion for information flow in the time dimension. The model inputs ($x_{s,t}$) at site $s$ and time $t$ are mapped to the output $\hat{y}_{s,t}$.

sequential models: A simple LSTM architecture, a model able to learn temporal dynamics via its built-in memory processing mechanism, and a TCN model, which applies 1D convolutions in time. Those sequential layers were stacked to achieve the extraction of complex temporal features. While the LSTM has, conceptually, an unlimited receptive field, the temporal context considered by the TCN depends on its hyperparameters. The neural network-based models use the building blocks illustrated in Fig. 3 and were implemented in PyTorch (Paszke et al., 2019) v1.13.

### 3.3 Model training

To identify models with the capacity to generalize well to unseen sites, we trained them following an eight-fold cross-validation scheme, for which the data splitting between sites was kept identical across different models and covariate setups. To decrease the dependency between the sets, we ensure that sites in close spatial proximity (below $0.05°$ distance) are part of the same set using clustering of coordinates. The site groups are provided in the Appendix (Tab. B1). For each of the eight folds, six of the cross-validation sets were used for training (75%), one for validation (12.5%), and one for testing (12.5%), such that each site appeared in the testing set once. The training and validation sets were used for model tuning with the early stopping algorithm: The model parameters were optimized on the training set, while the validation set was used to evaluate the generalizability regularly (ten times in each training epoch). Once the validation loss converged over a given number of validation steps (the "patience"), model training was halted, and the best parameters were restored. With these parameters, the model was applied to the independent test set. This approach yielded independent predictions for each site, which we then used to evaluate the model's performance on a site-level basis. For a speedup of the training, the model was iteratively fed with randomly selected sequences of two years. The first year was used for providing temporal context similar to the "spinup" in dynamic process models, while the second was used for tuning. Note that the two years were randomly sampled in every epoch, ensuring that all observations were, potentially, used for training with high likelihood.

We used a random search over a predefined set of hyperparameters. For each model, 20 parameter sets were sampled uniformly with replacement. The sets are reported in Table A1 in the Appendix. Note that we selected hyperparameter ranges based on prior experiments, i.e., we excluded values that performed consistently badly in order to obtain a denser sampling of the sensitive ranges. With this protocol, we tuned hyperparameters independently for each model except for `linearreg`, which has no hyperparameters. The same hyperparameter set was used throughout the cross-validation for each model setup. Thus, the cross-validation ensemble is composed of models with the same hyperparameters, but trained on different subsets of the data.

To quantify model uncertainty on the site level, we performed the cross validation for the six best-performing models (in terms of validation MSE) from hyperparameter tuning for each model and setup individually. For later analysis and upscaling, only the best-performing model was used.

## 3.4 Upscaling

To achieve global coverage, we fed the models with harmonized and gridded data from 2001 to 2021 with $0.05°$ spatial and hourly temporal resolution. Due to the high computational demands of the upscaling, we decided to use only two well-performing covariate setups across all models. We selected the `met+rs` and the `met+prec+rs+pft` (`full`) setups as they were among the best-performing setups. The `linearreg` model was excluded, as it performed significantly worse than the non-linear algorithms in the site-level cross-validation. For each of the four remaining ML models, we compute an ensemble of eight upscaling products. The members, hereafter referred to as "cross-validation ensemble", correspond to the models obtained from the cross-validation folds, i.e., each fold yielded one model which was trained and evaluated on an independent set of sites. Note that this differs from the X-BASE setup (Nelson & Walther et al., 2024), where the cross-validation was used exclusively for model evaluation, and the upscaling was done with a single model trained again on additional sites without holding out a test set. This method does not yield an ensemble, and is, therefore, not suited for the evaluation of within-model upscaling robustness. The upscaled products are then evaluated by a ML model inter-comparison and by contrasting global yearly sums and regional cross-validation ensemble mean and variability to independent products.

## 4 Results and discussion

### 4.1 Site-level modeling of evapotranspiration (ET)

In this section, the EC site-level prediction of ET is evaluated based on the cross-validation setup. We aim to understand the impact of different covariate types and ML approaches on performance at different temporal scales and assess the relevance of sequential model architectures on reproducing ET observed at EC sites.

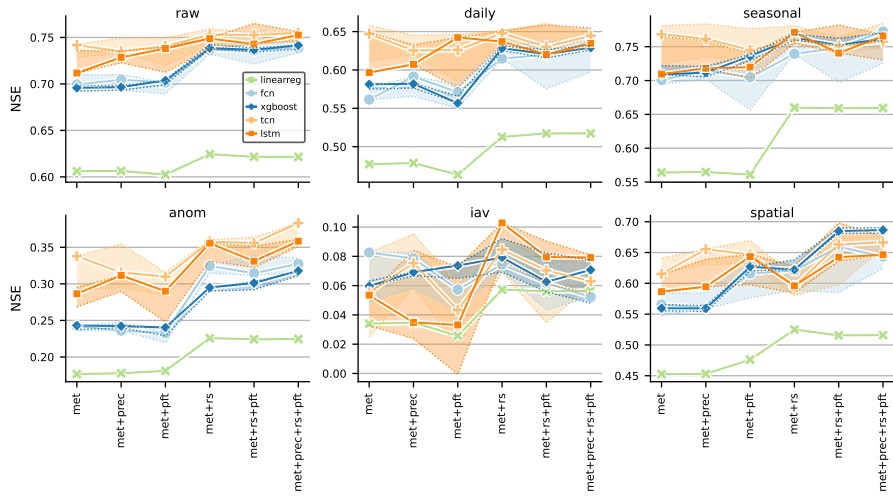

**Figure 4.** Site-level evaluation for modeled evapotranspiration: Median Nash–Sutcliffe model efficiency (NSE) across sites for different models (lines) and covariates (x-axis) for various scales (panels). The shaded area represents the range of the top six models from hyperparameter tuning per model setup, with the solid line indicating the best-performing model (top one). This can be interpreted as model uncertainty. Note that the model selection during hyperparameter tuning was based on the validation set and the mean squared error. Consequently, the best model is not necessarily the top performer in terms of NSE. The scales shown are `raw` for hourly, `daily` for daily aggregates, `seasonal` for daily seasonal, `anom` for daily anomalies (`daily` minus `seasonal`), and `iav` for interannual variability. The `spatial` scale reflects the NSE across site mean values, indicating the ability of the models to capture spatial variability. For certain temporal scales, some sites had to be excluded due to missing or infinite values; the number of sites used is indicated in the respective panel title.

### 4.1.1 Model performances across scales

Figure 4 presents site-level ET accuracy in terms of NSE for different ML models and covariate groups across scales. We now focus on the best-performing models (solid lines in Fig. 4). Overall, model outcomes were more influenced by the choice of covariates than by the ML algorithm used. Notably, a significant interaction between ML models and covariates was observed.

In general, the ML models outperformed linear regression by a substantial margin. On the `raw` and `daily` time scales, sequential models exhibited the best performance, maintaining stable NSE values of 0.70-0.75 (`raw`) and 0.60-0.65 (`daily`)

across data setups. Non-sequential models performed worse when using only meteorological covariates but showed significant improvement when remote sensing covariates were included, achieving similar performance to sequential models.

On the seasonal scale, sequential models outperformed others in the absence of remote sensing covariates. However, when remote sensing covariates were included, the performance differences between models became less pronounced. For anomalies, adding remote sensing covariates enhanced model performance across all setups, with sequential models consistently

outperforming non-sequential models. Notably, this was the only scale where precipitation had a clear positive impact across all models compared to the `met+rs+pft` setup.

For interannual variability, model performance was generally poor across all setups, with no clear patterns, and the small y-axis range underscores this. Adding PFTs and precipitation did not improve, and on the contrary in some cases, reduced performance. On the spatial scale, model performance was similar across models, although sequential models exhibited a slight advantage with the `met` and `met+prec` setups. Including PFT covariates notably improved performance for all models on this scale.

Regarding model uncertainty (shaded areas in Fig. 4), sequential models generally showed lower robustness than non-sequential models, likely due to their added complexity. Sequential models consistently outperformed non-sequential ones on the `raw` scale. On the `daily` and `seasonal` scales, the differences between sequential models were primarily driven by model uncertainty, i.e., the uncertainty ranges displayed in Fig. 4 overlapped strongly on the spatial scale. On the anomaly scale, sequential models reliably outperformed non-sequential models across the top six setups. Given the small performance range and large model uncertainties on the `iav` scale, caution is needed when interpreting these differences. Similarly, the differences on the spatial scale were minimal, and the relative uncertainties were large.

Next, we discuss the broader implications of these findings. The linear models (`linearreg`) consistently underperformed because evapotranspiration is governed by complex interactions and non-linear functions, making the advantages of ML methods particularly evident. While sequential models showed only marginal improvement when adding covariates related to ecosystem state, non-sequential models exhibited more substantial improvements. This serves as a sanity check for the sequential models: they were able to extract additional information from the temporal meteorological covariates, as expected. However, incorporating remote sensing covariates improved and stabilized their performance. Conversely, this suggests that remote sensing covariates serve as useful proxies for ecological memory: while sequential models could extract additional information from antecedent covariates, most of this information appeared to be contained in the remote sensing covariates, allowing non-sequential models to achieve similar performance.

On the anomaly scale, we observed a more noticeable performance increase when considering remote sensing even for the sequential models. This is important, as anomalies are crucial for studying and quantifying ecosystem responses to uncommon or extreme conditions. This performance increase could be linked to processes observable by remote sensing but not directly derivable from meteorology, such as the effect of management on crops and forests and natural disturbances. Here, adding precipitation improved the performance of all models. This is not surprising, as precipitation is a key driver of ET through its influence on soil moisture (Nelson et al., 2020). However, the improvement was comparatively small, which may be due to the use of precipitation reanalysis data that may not fully represent local conditions. The low performance on the `iav` scale was also reported by Jung et al. (2019) and Nelson & Walther et al. (2024).

It is worth noting that adding PFTs as covariates did not improve, and in some cases even reduced, model performance on the temporal scales. On the spatial scale, however, their inclusion was beneficial, highlighting the potential importance of spatial covariates for modeling EC fluxes and upscaling. PFTs have long been criticized for not accurately representing the continuous characteristics of ecosystems (Reichstein et al., 2014; Kattge et al., 2011). Our experiments suggest that while adding PFTs provides additional information for representing spatial patterns, they may also harm extrapolation on other scales, arguably due to the inflated covariate space; indeed, each of the nine PFTs introduces another input dimension due to one-hot encoding.

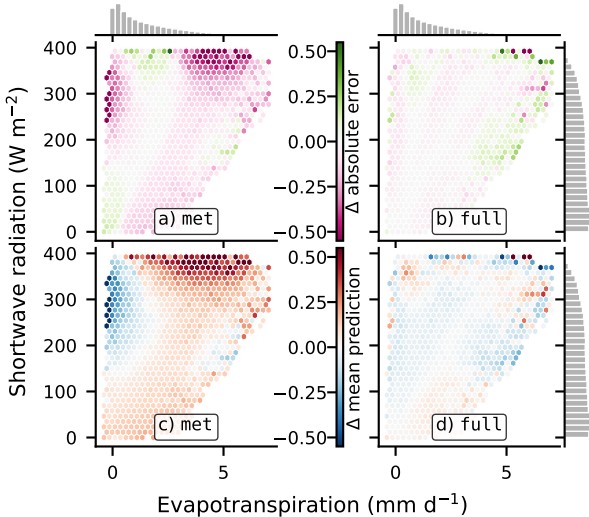

**Figure 5.** Comparison of a sequential (`lstm`) and a non-sequential (`xgboost`) model in terms of absolute error and mean predicted ET in the space of observed evapotranspiration × shortwave irradiation: Panel **a)** and **b)** show the *difference in absolute error* between the `lstm` and `xgboost` models, with **a)** showing the difference when only meteorological covariates are used, and **b)** showing the difference when precipitation, remote sensing, and PFT covariates (`full` setup) are included. Here, magenta indicates cases where the sequential model outperforms the non-sequential model, and green vice-versa. The bottom panels **c)** and **d)** show the *difference in mean predicted ET* between the `lstm` and `xgboost` models for the respective covariate setups. In these panels, red colors indicate an underestimation of ET by `xgboost` compared to `lstm`, and blue indicates the opposite. The histograms represent the marginal data distribution.

We therefore recommend exploring alternative spatially continuous variables, such as soil properties or plant traits, that could summarize ecosystem functional properties.

These findings advocate for comprehensive feature selection to identify more relevant static features, thereby avoiding un-
necessary inflation of the input dimensionality. Alternatively, or in addition, location embeddings, such as SatCLIP (Klemmer et al., 2024), could improve model generalizability by providing a condensed representation of land surface characteristics.

### 4.1.2   Memory effects matter

As previously mentioned, the performance gap between the non-sequential and sequential models decreased when remote sensing observations were incorporated as covariates. We explore these differences in Fig. 5: As shown in the top-left panel
(Fig. 5a), which displays the absolute error difference between the `lstm` and `xgboost` models, non-sequential models performed worse when high incoming radiation ($> 200 \ \mathrm{Wm^{-2}}$) was paired with either low or high observed ET. To represent these conditions, the models must implicitly learn about water availability. It appears that the sequential model was able to learn proxies of wetness from the meteorological time series, whereas the non-sequential model was unable to do so. Instead, the non-sequential model seems to have learned an average behavior, which performed well in most situations.

When remote sensing covariates were added as well as PFTs and precipitation, the differences in performance were reduced but not entirely eliminated (Fig. 5b). This interpretation is further supported by the bottom panels, Fig. 5c-d, which show the difference in mean predicted ET between the sequential and non-sequential models. With access to only meteorological covariates (Fig. 5c), the non-sequential model overestimated ET with high incoming radiation but low observed ET—representing dry conditions that the model failed to recognize. In contrast, high observed ET was underestimated by the non-sequential model,
which likely corresponds to wet conditions. After incorporating remote sensing, PFTs, and precipitation covariates (Fig. 5d), the performance differences were substantially reduced. This comparison underscores the importance of memory effects in ET modeling and illustrates how remote sensing covariates, while useful, are not perfect proxies for ecological memory.

    Furthermore, including precipitation may be beneficial for the sequential models in principle, as these models could learn soil moisture dynamics. However, as noted earlier, the performance increase was not as pronounced as expected (Fig. 4),
possibly due to the use of reanalysis data that may not fully capture local conditions.

## 4.2   Scaling evapotranspiration to global coverage

With the models optimized at the site level, we create global ensembles of ET estimates for the `met+rs` and the `full` (`met+prec+rs+pft`) covariate setups. The ensemble members were trained with different subsets of the training data within the cross-validation scheme. At the site level, the differences between ML models were small when considering remotely sensed
observations or PFTs as covariates. However, when scaling globally, data distribution shifts can (and will) affect different model types in different ways. These shifts arise from the different scales of the measurements (point at EC site versus grid globally), the different data products used (direct observation of meteorological variables at EC site versus reanalysis globally), and the spatial extrapolation into different ecoclimatological conditions from irregularly and sparsely sampled locations. In this section, we consider the performance of the different ML approaches and covariate setups while scaling out of the EC station
locations.

### 4.2.1   Global patterns of evapotranspiration

Figure 6 shows global annual ET estimates for all ML models and the two selected covariate setups. With the `met+rs` setup, non-sequential models (`xgboost`, `fcn`) estimated about $65 \cdot 10^3$ km$^3$ y$^{-1}$ global annual ET, which is close to the land surface models (`lsm`) ensemble mean ($64.7 \pm 6.9 \cdot 10^3$ km$^3$ y$^{-1}$), while sequential models (`tcn`, `lstm`) predicted roughly 8% lower
values (59.8 and $61.8 \cdot 10^3$ km$^3$ y$^{-1}$, respectively). With the `full` setup, `xgboost` showed the strongest increase in global ET (+10%), while `fcn`, `lstm`, and `tcn` increased modestly (1%, 0.5%, and 5%, respectively). `xgboost` was closest to `gleam` and `xbase`, whereas neural networks aligned more with `lsm` estimates.

    Figure 7 displays spatial patterns of ensemble mean ET per model and covariate setup. The models aligned well spatially in the `met+rs` setup (Fig. 7a), with lower ET by sequential models especially in regions in the Southern Hemisphere which
have sparse coverage by EC stations. The `full` setup (Fig. 7b) revealed stronger divergences across models, particularly in arid zones such as the Sahara and Arabian Peninsula and generally in the subtropic zones. Figure 7c shows that changes in ET

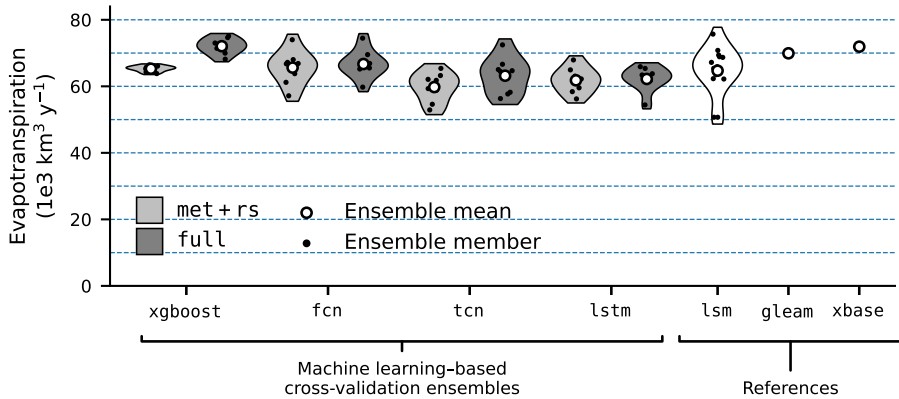

**Figure 6.** Global annual evapotranspiration (ET) per model. The violin plots represent the density of independent cross-validation runs (black dots), with their mean values across runs displayed as white dots. For the machine learning models, the `met+rs` setup (meteorological and remote sensing covariates) is shown in light grey, while the `full` setup (meteorological, precipitation, remote sensing, and PFT covariates) is shown in dark grey. The data from a number of land surface models (`lsm`), the GLEAM product (`gleam`), and FLUXCOM X-Base (`xbase`) are added as reference.

estimates due to covariate setup differences were low in temperate zones but more pronounced in arid and tropical regions. The `lstm` appeared relatively robust across setups.

A temporal anomaly correlation analysis (Fig. 8) reveals high agreement ($r > 0.8$) between covariate setups in most areas, except for arid zones, and only the `tcn` maintained high consistency globally. A comparison with `gleam` (Fig. 8b) showed moderate correlations (mean $r \approx 0.5$), with discrepancies mainly in tropical and arid regions. Figure 8c highlights where models improved alignment with GLEAM when using additional covariates: `tcn` improved slightly, while `lstm` improved most consistently across dry regions.

Figure 9 shows uncertainty via median absolute deviation (MAD) across ensemble members based on monthly values. `xgboost` had the lowest spread overall, with modest increases in tropical and more pronounced increase in arid zones in the `full` setup. The neural networks showed higher ensemble spread: in the tropics and subtropics for the `met+rs`, and in arid regions for the `full` setup. In the latter, uncertainty patterns aligned with ecosystem boundaries, likely reflecting PFT transitions.

We saw that models showed biases in terms of global ET sums, but apart from `xgboost`, the values were relatively robust to changes on covariate sets. From the spatial and temporal patterns (Fig. 7, Fig. 8, and Fig. 9), it becomes evident that the uncertainty originates mainly from tropical and arid to semiarid regions. Finally, we want to investigate if the differences are related to the technical setup (architecture and covariates) or to the variability in training data used in cross validation. Therefore, we investigate the alignment of members shown in Fig. 6: We compute the linear ($r$) and rank ($\rho$) correlation between global sums of ET between all models and covariate setups. In other words, we quantify if using the same data subset in cross validation leads to a consistent upscaling behavior in terms of global ET. The results are shown in Table 2. Global ET estimates

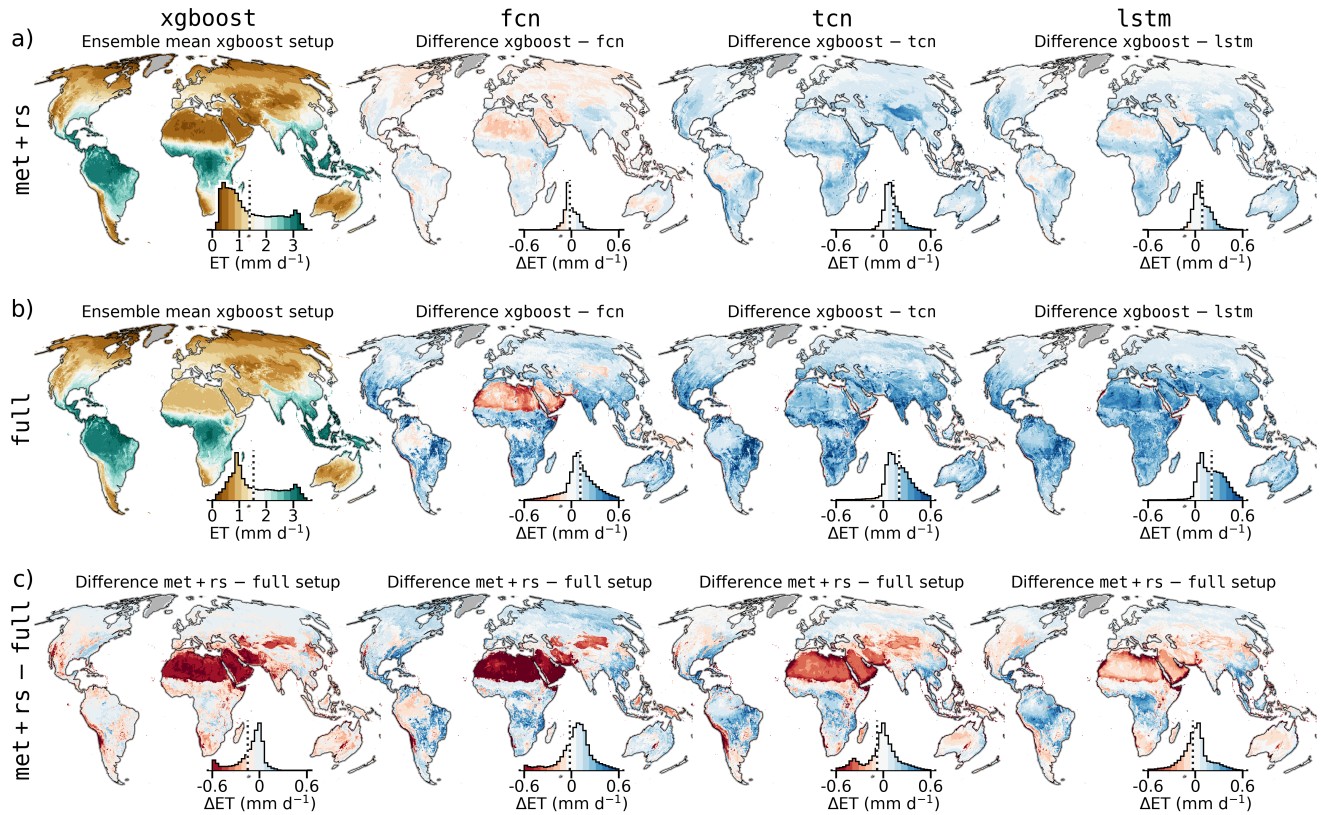

**Figure 7.** Spatial model evaluation and comparison for the `met+rs` and `full` (meteorological, precipitation, remote sensing, and PFT covariates) setup. Grid-level cross-validation ensemble mean ET for each model is shown in the top row (**a**)) for `met+rs`, and the middle row (**b**)) for `full`. The first column represents `xgboost`, and the remaining columns show the difference to `xgboost`. The bottom row (**c**)) shows the difference `met+rs` minus `full`. The color scale is consistent across panels and reflects mean ET in $\mathrm{mm\,d^{-1}}$. Inset histograms show the distribution of values weighted by grid cell area; the median is indicated by a dashed black line.

from `xgboost` were highly correlated between setups ($r = 0.90$, $\rho = 0.76$), suggesting consistent behavior across training data subsets. Neural networks exhibited lower correlation ($r = 0.32$–$0.75$), indicating greater variability. Overall, consistency in upscaling behavior was not strongly tied to model type or covariate setup.

### 4.2.2 Model biases across architectures and covariate setups

Most models, particularly the sequential ones, estimated global ET—averaged across members—at the lower end of current estimates from independent methods of about $70 \pm 5 \cdot 10^3 \ \mathrm{km^3\,y^{-1}}$ (Jung et al., 2019), and compared to `gleam` and `xbase`. Energy-balance non-closure at EC sites could depress ET by up to 20% (Jung et al., 2019), yet recent findings suggest that the closure gap may contribute only moderately to the global ET bias, as the closure gap very likely stems primarily from sensible

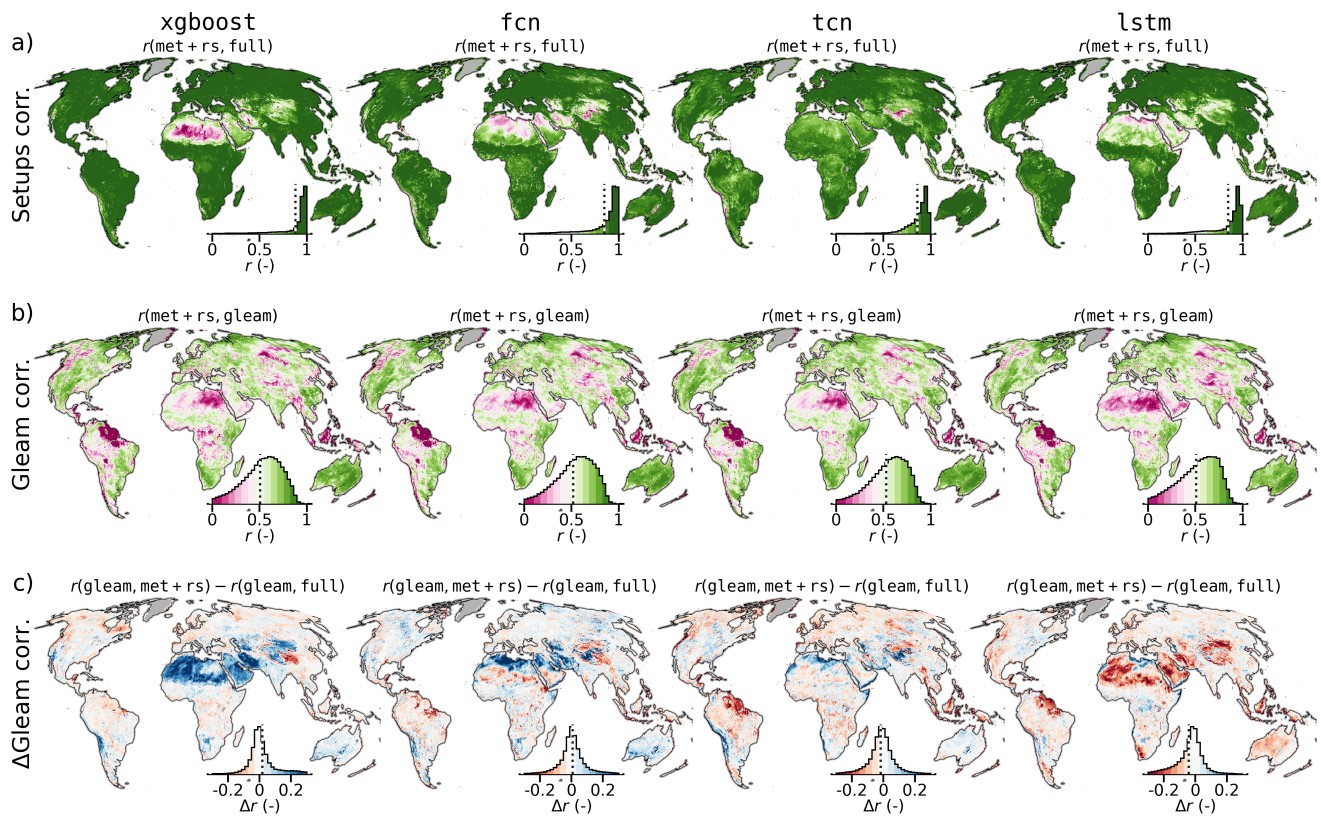

**Figure 8.** Temporal correlation of monthly ET anomalies among models and covariate setups. **a)** Correlation between monthly ET anomaly time series from `met+rs` and `full` setups. **b)** Correlation between monthly ET anomalies of `met+rs` and the GLEAM product. **c)** Difference in temporal correlation with GLEAM between the `full` and `met+rs` setups. Inset histograms show area-weighted value distributions with dashed black lines indicating the median.

rather than latent heat fluxes, i.e., ET (Zhang et al., 2024; Mauder et al., 2024). Because both `xgboost-full` and the `xbase`
product would share the same closure issue but still reach the benchmark range, the deficit of the other models must originate
elsewhere; options are training data subsets, the ML approaches, covariate setup, or data distribution shift due to extrapolation
into underconstrained regions and product types (site level versus gridded).

Disentangling these possible causes is difficult, but we can draw some conclusions from the results. We have shown that
different training subsets did not lead to consistent upscaling behavior (Table 2). In other words, taking the same training
data subset for model training did not lead to similar behavior across ML models in terms of global ET. This suggests that
the ensemble variance at global scale is not driven by training data, but rather by how the ML models extrapolate out of the
training data distribution. It indicates that, when evaluating the global sums alone, neither neural networks versus `xgboost`

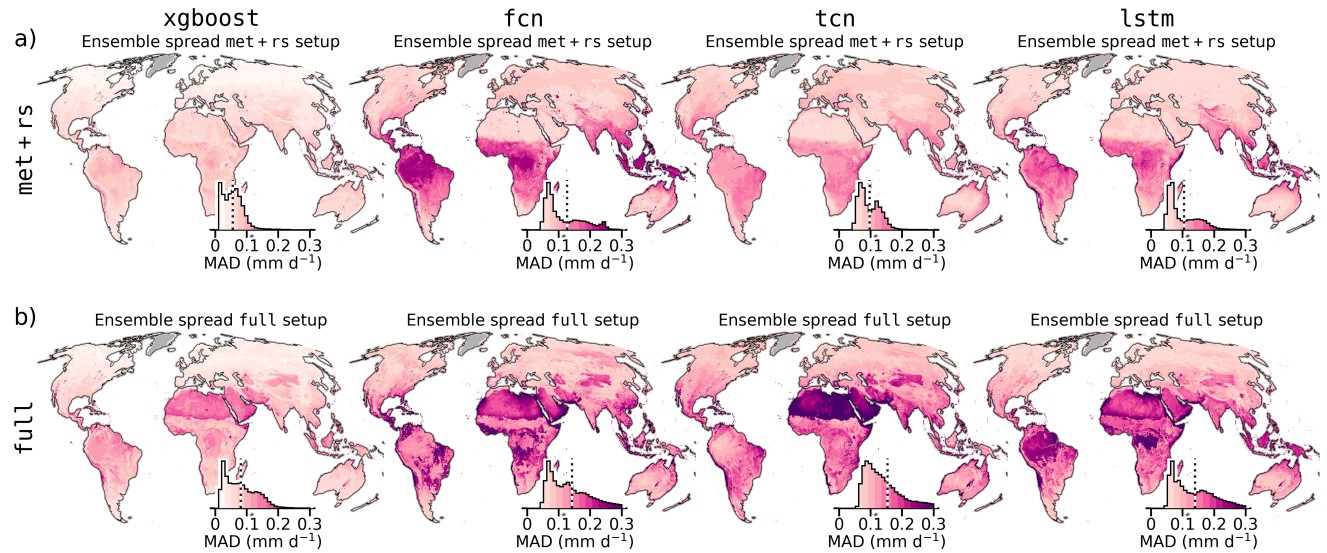

**Figure 9.** Median absolute deviation (MAD) of the cross-validation ensemble for each model setup, representing spatial uncertainty in $\text{mm}\,\text{d}^{-1}$. **a)** MAD for the `met+rs` setup. **b)** MAD for the `full` setup. Inset histograms show the distribution of values weighted by grid cell area; the dashed black line marks the median.

**Table 2.** Global evapotranspiration (ET) correlation across model ensemble members. The correlation is based on global ET sums across cross-validation members trained on the same sites. The upper triangle (bold) shows Pearson correlation, the lower triangle Spearman rank correlation.

|  |  | xgboost | | fcn | | tcn | | lstm | |
|---|---|---|---|---|---|---|---|---|---|
|  |  | rs | full | rs | full | rs | full | rs | full |
| xgboost | rs | 1.00 | **0.90** | **0.25** | **0.23** | **0.63** | **0.45** | **0.67** | **0.20** |
|  | full | 0.76 | 1.00 | **0.57** | **0.36** | **0.58** | **0.65** | **0.80** | **0.39** |
| fcn | rs | 0.12 | 0.71 | 1.00 | **0.58** | **0.16** | **0.43** | **0.44** | **0.78** |
|  | full | 0.26 | 0.62 | 0.52 | 1.00 | **0.20** | **0.25** | **-0.08** | **0.45** |
| tcn | rs | 0.33 | 0.38 | 0.29 | -0.05 | 1.00 | **0.75** | **0.39** | **0.08** |
|  | full | 0.40 | 0.81 | 0.81 | 0.38 | 0.62 | 1.00 | **0.64** | **0.09** |
| lstm | rs | 0.43 | 0.55 | 0.52 | -0.12 | 0.38 | 0.62 | 1.00 | **0.32** |
|  | full | -0.10 | 0.36 | 0.76 | 0.31 | 0.10 | 0.33 | 0.48 | 1.00 |

nor sequential versus non-sequential models can be regarded as inherently more consistent—or therefore more reliable—for global upscaling.

A further complication is the change from EC site observed meteorology to reanalysis grids. At EC sites, meteorological variables are locally observed, whereas global inputs are derived from reanalysis grids (with a spatial resolution of 0.25°) with assimilation and spatial smoothing (Hersbach et al., 2020; Parker, 2016; Grusson and Barron, 2022; Valmassoi et al., 2023). Sequential networks, which exploit fine-scale temporal structure, are theoretically more sensitive to such a distribution shift than tree-based or feed-forward models. Yet our results show that spatial patterns are robust in data-rich regions, both across

architectures (ensemble means, Fig. 7) and within architectures (ensemble spread, Fig. 9). Because every model and covariate set converges where observational support is strong, the station-to-grid shift cannot be the dominant source of the spatial or global-sum discrepancies. Instead, the residual differences arise from the extrapolation into data-scarce regions, with different behavior across models and covariate setups.

    The model runs without PFTs (Fig. 7a and Fig. 9a) showed a better agreement among each other (Fig. 7a vs. Fig. 7b)

and within ensemble members (Fig. 9a vs. Fig. 9b). This suggests that excluding the PFTs from the covariates improves the model robustness. However, we saw at site level that the PFTs improved the model performance particularly at the spatial scale (Fig. 4). At the same time, we observed unrealistic spatial patterns with the `full` setup: In particular, `xgboost` and `fcn` with the `full` setup estimated large ET in arid regions like the Sahara and Arabian Peninsula. These large ET estimates align poorly with our understanding of ET processes, and `xbase` estimates in these areas have also been shown to be too high

(Nelson & Walther et al., 2024). Thus, while `fcn` and particularly `xgboost` with the `full` covariate setup appear close to the global mean ET benchmark, this is likely for the wrong reasons, i.e., overestimation in arid regions. The sequential models also yielded higher global ET when including PFTs, likewise originating from arid regions, but the increase was less pronounced. The `lstm` showed the largest robustness across setups. For all models, we observed an increase of ensemble spread in arid and also tropical regions when adding PFTs (Fig. 9). It is possible that in the site-level cross validation, these effects were not

visible because of a lack of EC stations in arid and tropical regions.

    While we cannot answer which model in combination with which covariate setup yields the "best" upscaling results, we can conclude that non-temporal models yielded more realistic global ET estimates, even if this is largely due to covariate-driven artifacts in data-scarce regions rather than genuinely improved process representation. The sequential models were also sensitive to covariate setup, but the `lstm` was more robust compared to the other models.

### 4.2.3   Consistent temporal dynamics

Despite variability in absolute ET magnitudes, the models showed strong internal consistency in their temporal dynamics (Fig. 8). The high correlation between covariate setups (Fig. 8a) indicates that models captured similar temporal variations and responded consistently to climatic forcing.

    Temporal correlation with `gleam` was moderate overall, as discussed previously. The GLEAM ET product is based on con-

415 ceptual models driven by remote sensing inputs and meteorological forcing. We do not treat it as ground truth, but we assume it provides spatially consistent temporal patterns due to its incorporation of prior knowledge. In contrast, our ML models rely solely on data-driven representations, and may be sensitive to data shifts due to generalization beyond data support and data types. Given this, the overall moderate agreement with GLEAM is not surprising and does not imply poor model performance.

Rather, we interpret divergences—especially in tropical and arid regions—as a consequence of low data availability and known challenges in these environments.

Temporal correlation with `gleam` improved notably for the `lstm` model in arid and some tropical regions when additional covariates were included (Fig. 8c). This improvement was not observed for `xgboost`, which showed a slight degradation, particularly in arid climates, nor for `fcn`, which remained largely unchanged. We attribute the improved performance of sequential models primarily to the inclusion of precipitation, consistent with their enhanced performance on anomaly scale at site level when this covariate was added (Fig. 4).

It is unclear why `tcn` did not show a similar improvement in arid regions. At site level (Fig. 5), `tcn` actually outperformed `lstm` on the anomaly scale, albeit with higher model uncertainty. This could suggest that `lstm` learns a more robust representation of temporal dynamics in these regions, which is not evident from the site-level cross-validation as those regions are largely absent from the training data. In contrast, `tcn` may be more sensitive to distribution shifts. There is evidence that `lstm` architectures are better suited for capturing hydrologically relevant temporal patterns compared to convolutional models (Kraft et al., 2025). While the `lstm` can theoretically access the full temporal context, `tcn` is constrained to a fixed temporal window of approximately 8 days in both the `met+rs` and `full` setups (Tab. A1). Note that this temporal context is, for the `tcn` architecture, related to the hyperparameters. This limited context may explain its reduced ability to generalize temporal patterns in data-sparse regions.

In summary, temporal dynamics remained relatively consistent across methods. Sequential models, especially `lstm`, benefited from richer covariate input, particularly precipitation, improving alignment with an independent product in arid regions. This highlights the added value of using sequential models for representing temporal dynamics in ET modeling and upscaling, yet the interaction between model architectures and covariates, particularly in data-sparse regions, also suggests that a profound covariate selection is necessary to identify best-working setups.

## 4.3 Lessons learned and outlook

### 4.3.1 Site-level performance: small but consistent advantages for sequential models

At the site level, sequential models consistently outperformed non-sequential models for ET flux prediction, especially when evaluated on the anomaly scale (Fig. 4). This is in line with previous findings for NEE modeling (Besnard et al., 2019), suggesting that temporal dependencies, such as ecosystem memory effects, are captured more effectively by models with recurrent or convolutional memory. The inclusion of remote sensing variables (e.g., vegetation indices) reduced the performance gap between sequential and non-sequential models by providing state proxies, although the advantage for sequential models persisted. Adding precipitation from reanalysis further improved performance at the anomaly scale, particularly for the sequential models, confirming the value of dynamic hydrological information. However, these improvements were relatively modest, indicating that data availability and quality, rather than model architecture, remain the primary bottlenecks. More advanced architectures, such as temporal transformers, may offer further gains, but recent work in GPP modeling has shown only marginal improvements (Nakagawa et al., 2023), providing further evidence that EC flux modeling is still a data-limited problem.

### 4.3.2 Global upscaling: both model architecture and covariates drive uncertainty

Small changes in the covariate setup led to moderate differences in global ET means (Fig. 6), especially for `xgboost` with the `full` setup, which showed an increase of 10% in global ET, supposedly due to strong leverage effects of PFTs in sparsely sampled regions. The moderate differences were partially due to error compensation. While we found no clear evidence that any architecture yields more realistic absolute ET estimates, our results point to systematic biases that depend on both model design and the covariates used. This highlights the importance of methodological choices, including architecture, covariate selection, and cross-validation design, in driving upscaling uncertainty.

### 4.3.3 Spatio-temporal patterns: robust dynamics, variable magnitudes

Despite differences in magnitude, all models showed consistent temporal ET patterns and strong alignment in anomaly correlations, both among themselves and, to a lower extent, also with the GLEAM product (Fig. 8), yet the robustness drastically decreased outside of the training domain. With the inclusion of precipitation and PFTs in the `full` setup, the sequential models—particularly `lstm`—showed improved temporal agreement with GLEAM in arid regions, supporting their value for representing soil moisture-related memory effects. However, these benefits were largely limited to the temporal dimension; biases in global means and ensemble spread remained pronounced, especially for data-sparse regions where additional covariates such as PFTs introduce high leverage. Given their lower computational complexity and more stable behavior, we find that non-sequential models like `xgboost` currently offer a pragmatic and robust solution for global-scale ET upscaling, especially when paired with carefully selected meteorological and remote sensing inputs.

### 4.3.4 Outlook: addressing biases and uncertainties in data-scarce regions

Moving forward, reducing model biases and uncertainty in upscaling requires both methodological advances and improved data. Rather than prioritizing complex model architectures, future efforts should focus on four complementary pathways:

**i) Feature selection and additional data constraints:** Deep learning presents promise for enhancing flux modeling and upscaling, and offers advanced computational techniques capable of managing complex, non-linear interactions within ecosystems. However, to maximize the effectiveness of deep learning in such a data-limited setting, it is essential to implement additional constraints and integrate richer data sources (Reichstein et al., 2019). The accuracy of deep learning models heavily relies on the quality and diversity of the input data (Karpatne et al., 2019). Enhancing these models with additional covariates that accurately reflect ecological and atmospheric conditions can significantly improve their predictive power. Additionally, expanding the network of flux stations and sharing the data for scientific applications would enhance the data base to cover more diverse ecological conditions and climate zones, thereby enriching the training data used for model calibration and validation. Furthermore, applying constraints at a regional level, akin to the approach by Upton et al. (2024), who used an ensemble of atmospheric inversions of NEE as large-scale guidance for flux upscaling, could be used to reduce biases. For ET modeling, large-scale water balance could be used as a regional constraint, for example.

**ii) Additional data sources:** To further refine the performance of deep learning approaches in EC measurement upscaling, it could be beneficial to tap additional data sources. Approaches such as transfer learning can be particularly effective (Caruana, 1997; Pan and Yang, 2010). By applying knowledge gained from one region to another, or from related and richer datasets, models can achieve better generalization, especially in data-sparse areas. To deal with shifts in covariates due to the various reasons discussed previously, domain adaptation (He et al., 2023) could provide a useful toolbox to reduce upscaling biases.

**iii) Physical constraints:** As a complementary pathway, incorporating prior scientific knowledge into deep learning models could help address challenges associated with data extrapolation and distribution shifts encountered in upscaling (Reichstein et al., 2019; Kraft et al., 2022). Such integration aids in aligning model outputs with established physical laws and ecological principles, thereby improving the reliability of the predictions (Reichstein et al., 2022). Physics-informed and hybrid physics/ML approaches represent a cutting-edge direction in the field of flux modeling, as they merge the empirical strengths of deep learning with the deterministic nature of physical models. For upscaling into undersampled regions, such constraints can nudge the model outputs towards physically more plausible solutions. As an example, encoding simple relationships between precipitation and evaporation, or vegetation and transpiration, could help reducing ET estimates in arid regions, where EC stations are lacking. Although challenging, more comprehensive physical process parameterizations, such as the Penman-Monteith equations, can be combined with machine learning to estimate ET (Zhao et al., 2019; ElGhawi et al., 2023). This could, in principle, reduce the widely reported regional biases in upscaling EC fluxes with machine learning, which we identified as currently the main challenge in flux upscaling.

**iv) Systematic benchmarking:** Finally, systematic benchmarking frameworks, such as FLUXCOM-X (Nelson & Walther et al., 2024), are essential for disentangling the complex interactions between models, covariates, and evaluation setups. Such frameworks enable controlled ablation studies and targeted diagnostics, helping to build a more transparent understanding of uncertainty sources in ML-based flux upscaling.

In conclusion, while deep learning provides valuable tools for modeling land-atmosphere interactions, the key to better global ET estimates lies in more comprehensive observational data, thoughtful covariate selection, targeted physical constraints, and methods to reduce extrapolation bias—rather than in model complexity alone.

## 5  Conclusions

In this study, we assessed different covariate setups and machine learning approaches for modeling ET fluxes at eddy covariance sites through cross-validation, and evaluated the robustness and quality of globally upscaled ET estimates. From our site-level analysis, we conclude that sequential deep learning models can outperform non-sequential models for ET flux modeling, particularly on the anomaly scale. The sequential models captured memory effects related to water availability, which improved their ability to represent temporal dynamics of ET. However, this advantage diminished when remote sensing covariates were included, as these effectively act as proxies for ecosystem memory. Adding precipitation as a covariate led to small performance improvements, especially for sequential models and on the anomaly scale, confirming its value in representing hydrological constraints. The inclusion of PFTs increased the ability of the models to capture spatial variability across sites, yet their discrete nature raises concerns about their broader utility. We, therefore, recommend further exploration of alternative static variables or targeted feature selection to maintain a parsimonious covariate set.

Globally, sequential and non-sequential models produced small but systematic differences in mean ET, suggesting that different model types learn distinct representations and respond differently and randomly when extrapolating out of the training distribution. When using PFTs as additional covariate, both divergence of spatial means of ET among models and ensemble spread within models increased, particularly in arid regions where data support is limited. This highlights the large leverage of the PFTs on upscaling results when extrapolating out of the covariate space, and underscores the need for better understanding of the role of covariates in upscaling. While the sequential models suffered from similar biases as their non-sequential counterparts, they achieved better temporal alignment with the GLEAM product, particularly when precipitation was included.

In the context of our experiments, we can answer the research questions posed in the introduction as follows:

a) *Do the sequential models lead to more realistic and robust global and regional ET estimates?*

No, sequential models yielded lower global ET estimates than expected from independent estimates. However, the larger estimates of the non-sequential models were likely due to an overestimation of ET in arid regions. The sequential models, especially the LSTM, were more robust to covariate shifts, though, supposedly due to their lower dependency on PFT due to their ability to link spatial variability to temporal features in the covariates.

b) *Do the sequential models capture the temporal dynamics of ET better?*

Yes, the sequential models captured temporal dynamics better than non-sequential models, at both site and upscaling level, particularly when using precipitation as additional input and with the LSTM architecture.

c) *How stable are the sequential models to the covariate setup and training data subsets, compared to the non-sequential models?*

The sequential models showed a similar ensemble spread than the non sequential neural network, indicating that robustness towards training data subsets is rather linked to the model type (decision tree versus neural network) than to the sequential nature of the models.

Considering the added complexity and computational cost of sequential neural networks, the relatively modest performance gain at site level, and their underestimation of global ET alongside larger ensemble spreads, we conclude that these models are not strictly required for global ET upscaling. Non-sequential machine learning approaches, such as XGBoost, can deliver similarly robust estimates across scales when supported by high-quality meteorological and remote sensing covariates. Yet, XGBoost overestimated ET in arid regions, which raises some concerns, but these are also not exclusive to this model type. Particularly the LSTM architecture was robust towards covariate setups and able to learn subtle temporal dynamics, which could be beneficial in specific applications. These findings highlight the importance of structural model diversity in ensemble setups to assess the robustness and uncertainty of upscaling results.

The potential for upscaling ET to a global scale using modern ML methods is constrained by the information content and representativeness of the EC training data. As such, seemingly small changes in covariate setups can exert large influence over upscaling results, particularly in data-sparse regions. To enhance the robustness and physical consistency of global ET products, we recommend pursuing complementary strategies: integrating richer covariates, increasing EC site coverage, applying regional constraints, leveraging transfer learning, and embedding prior scientific knowledge into ML architectures. By following these pathways, deep learning has the potential to produce more accurate, robust, and physically grounded predictions of terrestrial evapotranspiration across diverse environmental settings.

*Code availability.* TEXT

*Data availability.* TEXT

The upscaled ET fields generated for this study are available from the corresponding author upon reasonable request.

*Code and data availability.* TEXT

*Sample availability.* TEXT

*Video supplement.* TEXT

## Appendix A

Table A1 provides an overview of the hyperparameter search space and the best-performing combination of settings for each model and covariate setup. We sampled 20 random hyperparameter configurations for each model, evaluating them via early stopping on a validation set to determine the final selections. For the `tcn` model, the temporal context length considered (`temp. contex`) depends on the hyperparameters of the model, leading to a range of context lengths from about 4 days (91 hours) to about 19 days (451 hours). The number of parameters in each model (`# parameters`) was derived from the chosen architecture, highlighting that the complexity increases substantially with deeper layers or larger hidden sizes.

## Appendix B

Table B1 displays the eight cross-validation groups, each column representing one group, with its corresponding eddy covariance site IDs. We used this grouping to ensure that training and validation sets differ systematically across folds, minimizing spatial autocorrelation effects at the site level. The sites within each group span a range of climatic and ecophysiological conditions, allowing for more robust evaluation of model generalization.

*Author contributions.* BK implemented the model architectures and data pipeline for the neural networks and performed the analysis. He took the lead in writing the manuscript. JN, SW, FG, MJ, BK, and ZH contributed to the FLUXCOM-X framework, on which this study builds upon. SW, JN, FG, UW, and MJ provided, processed, and cleaned the datasets used. BK, JN, SW, MJ, GD, MRe, and WZ contributed to the scientific evaluation of the results. MK, MRu, and DT contributed with their expertise to the data-scientific and machine learning aspects of the study. All co-authors contributed to the manuscript.

*Competing interests.* The contact author has declared that none of the authors has any competing interests.

*Acknowledgements.* This research was funded by the European Research Council (ERC) Synergy Grant *Understanding and modeling the Earth System with Machine Learning (USMILE)* under the Horizon 2020 research and innovation program (Grant Agreement No. 855187). We thank the Max Planck Institute for Biogeochemistry for providing the computational and data infrastructure for conducting this research. We appreciate the financial support by the Max Planck Society to publish this manuscript open-access.

**Table A1.** Hyperparameter search space and the best performing hyperparameter per model and setup. The best combination was found by evaluating 20 random samples based on early stopping validation loss. The `xgboost` and `fcn` models are non-sequential, the `lstm` has minimum of one year (9k hourly time steps) and maximum of two years of theoretical context (18k hourly time steps), and the `tcn` model's temporal context depends on the hyperparameters (reported in the row `temp. context`), ranging from 90 to 450 hours, *i.e.*, about 4 to 19 days. The number of parameters per model is reported in the row `# parameters`.

| Model | Hyperparameter | Search space | Selected hyperparameter per setup | | | | | |
|-------|----------------|--------------|------|----------|---------|--------|----------------|---------------------|
| | | | met | met+ prec | met+ pft | met+ rs | met+ pft rs | met+ prec pft rs |
| xgboost | max_depth | $\{6, 8, 10, 12\}$ | 10 | 10 | 12 | 8 | 12 | 12 |
| | learning_rate | $\{10^{-2}, 10^{-1}, 2 \times 10^{-1}\}$ | $10^{-1}$ | $10^{-1}$ | $10^{-1}$ | $10^{-1}$ | $10^{-1}$ | $10^{-1}$ |
| | min_child_weight | $\{1, 5, 10\}$ | 1 | 1 | 5 | 5 | 5 | 5 |
| | max_delta_step | $\{1, 5, 10\}$ | 10 | 10 | 10 | 5 | 5 | 5 |
| | # parameters | derived | 209K | 212K | 640K | 128K | 1 264K | 1 273K |
| fcn | num_hidden | $\{128, 256\}$ | 128 | 128 | 128 | 256 | 256 | 128 |
| | num_layers | $\{3, 4\}$ | 1 | 1 | 1 | 2 | 1 | 1 |
| | dropout | $\{0.0, 0.2\}$ | 0.2 | 0.2 | 0.2 | 0.2 | 0.2 | 0.2 |
| | learning_rate | $\{10^{-6}, 10^{-5}, 10^{-4}\}$ | $10^{-6}$ | $10^{-4}$ | $10^{-6}$ | $10^{-5}$ | $10^{-4}$ | $10^{-4}$ |
| | weight_decay | $\{10^{-3}, 10^{-2}, 10^{-1}, 10^{0}\}$ | $10^{-3}$ | $10^{-1}$ | $10^{-3}$ | $10^{-3}$ | $10^{-1}$ | $10^{-1}$ |
| | # parameters | derived | 17K | 17K | 19K | 134K | 70K | 19K |
| tcn | num_hidden | $\{64, 128, 256\}$ | 64 | 256 | 256 | 256 | 256 | 256 |
| | num_layers | $\{2, 3, 4\}$ | 4 | 4 | 3 | 4 | 4 | 3 |
| | kernel_size | $\{4, 8, 16\}$ | 16 | 4 | 16 | 4 | 4 | 16 |
| | dropout | $\{0.0, 0.2\}$ | 0.2 | 0.2 | 0.2 | 0.2 | 0.2 | 0.2 |
| | learning_rate | $\{10^{-6}, 10^{-5}, 10^{-4}\}$ | $10^{-6}$ | $10^{-5}$ | $10^{-6}$ | $10^{-5}$ | $10^{-5}$ | $10^{-6}$ |
| | weight_decay | $\{10^{-3}, 10^{-2}, 10^{-1}\}$ | $10^{-3}$ | $10^{-2}$ | $10^{-2}$ | $10^{-2}$ | $10^{-2}$ | $10^{-1}$ |
| | temp. context | derived | 451 | 91 | 211 | 91 | 91 | 211 |
| | # parameters | derived | 473K | 2.0M | 5.6M | 2.0M | 2.2M | 5.6M |
| lstm | num_hidden | $\{64, 128, 256\}$ | 128 | 256 | 256 | 128 | 64 | 256 |
| | num_layers | $\{1, 2\}$ | 2 | 1 | 1 | 2 | 1 | 1 |
| | dropout | $\{0.0, 0.2\}$ | 0.0 | 0.2 | 0.2 | 0.2 | 0.0 | 2.0 |
| | learning_rate | $\{10^{-6}, 10^{-5}, 10^{-4}\}$ | $10^{-4}$ | $10^{-4}$ | $10^{-6}$ | $10^{-6}$ | $10^{-6}$ | $10^{-4}$ |
| | weight_decay | $\{10^{-3}, 10^{-2}, 10^{-1}\}$ | $10^{-1}$ | $10^{-1}$ | $10^{-3}$ | $10^{3}$ | $10^{-1}$ | $10^{-1}$ |
| | # parameters | derived | 217K | 336K | 344K | 219K | 25K | 348K |

**Table B1.** The eddy covariance site groups used for cross validation. The eight groups correspond to columns, the items correspond to site IDs.

| Group 1 | Group 2 | Group 3 | Group 4 | Group 5 | Group 6 | Group 7 | Group 8 |
|---------|---------|---------|---------|---------|---------|---------|---------|
| AR-Vir  | AR-TF1  | AU-Fog  | AU-DaP  | AU-ASM  | AU-Cum  | AR-SLu  | AU-Ade  |
| AT-Neu  | AU-Wac  | BE-Lcr  | AU-Whr  | AU-Dry  | AU-RDF  | AU-Cpr  | AU-DaS  |
| AU-Tum  | BR-Sa3  | CA-ER1  | CH-Cha  | AU-Emr  | AU-TTE  | AU-Gin  | BE-Maa  |
| BE-Dor  | CA-NS2  | CA-SF1  | CH-Fru  | BE-Bra  | AU-Wom  | AU-Rob  | CA-Qfo  |
| CA-NS3  | CA-NS5  | CA-SF3  | CN-Dan  | BE-Lon  | BE-Vie  | AU-Ync  | CH-Dav  |
| CA-NS4  | CA-TP1  | CN-Qia  | CN-Du2  | BR-Sa1  | CA-Cbo  | BR-Npw  | CZ-KrP  |
| CN-HaM  | CA-TPD  | DK-Sor  | CN-Du3  | CA-Obs  | CA-LP1  | CA-DB2  | CZ-Lnz  |
| CZ-BK1  | DE-RuR  | FR-Aur  | CN-Sw2  | CA-SF2  | CA-NS6  | CA-DBB  | DE-Hte  |
| CZ-BK2  | DE-RuS  | FR-Bil  | CZ-Stn  | CA-TP3  | CH-Lae  | CA-Gro  | DK-Gds  |
| CZ-RAJ  | DE-RuW  | FR-Fon  | DE-Kli  | CA-TP4  | CN-Cha  | CA-NS7  | ES-Agu  |
| DE-Akm  | DE-Seh  | FR-Tou  | DE-Lnf  | DE-Hai  | CN-Cng  | CA-Oas  | ES-LJu  |
| DE-HoH  | DE-Spw  | GF-Guy  | ES-Abr  | DE-Tha  | CN-Din  | CA-TP2  | ES-LM1  |
| DE-Lkb  | DE-Zrk  | GL-Dsk  | ES-LgS  | FI-Ken  | DE-Geb  | CH-Aws  | ES-LM2  |
| DE-Obe  | ES-Amo  | IE-Cra  | FI-Hyy  | FI-Lom  | FI-Sii  | CH-Oe1  | ES-Ln2  |
| DE-SfN  | FI-Var  | IT-BFt  | FI-Let  | FR-Lam  | GH-Ank  | CH-Oe2  | FI-Qvd  |
| DK-Eng  | FR-Hes  | IT-Cp2  | FR-EM2  | GL-NuF  | PA-SPn  | CZ-wet  | FR-FBn  |
| ES-Cnd  | IL-Yat  | IT-Cpz  | IT-Tor  | IT-Col  | PA-SPs  | DE-Gri  | FR-LBr  |
| FI-Jok  | IT-BCi  | JP-SMF  | MX-Tes  | IT-PT1  | SE-Lnn  | DE-Hzd  | FR-Pue  |
| GL-ZaF  | IT-Ro1  | US-Atq  | NL-Hor  | IT-SR2  | SE-Ros  | DK-Fou  | IT-Isp  |
| GL-ZaH  | IT-Ro2  | US-CS1  | PE-QFR  | IT-SRo  | US-A32  | FI-Sod  | IT-La2  |
| IT-Lsn  | NL-Loo  | US-CS2  | RU-Fy2  | JP-MBF  | US-Cop  | FR-Gri  | IT-Lav  |
| IT-MBo  | RU-Ha1  | US-CS3  | RU-Fyo  | MY-PSO  | US-EDN  | FR-LGt  | IT-Noe  |
| IT-Ren  | SD-Dem  | US-CS4  | SE-Htm  | RU-Cok  | US-Ho2  | IT-CA1  | RU-Che  |
| SE-Nor  | US-IB2  | US-GBT  | US-ARb  | SJ-Blv  | US-Me1  | IT-CA2  | SE-Svb  |
| SJ-Adv  | US-KS1  | US-GLE  | US-ARc  | US-CRT  | US-Me3  | IT-CA3  | US-Blo  |
| US-AR2  | US-Los  | US-Ha1  | US-CF1  | US-ORv  | US-Me6  | SE-Deg  | US-Ivo  |
| US-ARM  | US-MOz  | US-KS2  | US-CF2  | US-Prr  | US-Myb  | SN-Dhr  | US-KFS  |
| US-BZB  | US-NR1  | US-KS3  | US-CF3  | US-Tw1  | US-Ne1  | US-AR1  | US-UMB  |
| US-BZF  | US-ONA  | US-Lin  | US-CF4  | US-Tw2  | US-Ne2  | US-Bi1  | US-UMd  |
| US-BZS  | US-Ro1  | US-Me2  | US-Jo2  | US-Tw3  | US-Ne3  | US-Bi2  | US-Wi0  |
| US-BZo  | US-Ro4  | US-Me5  | US-NGB  | US-Tw4  | US-Sne  | US-Goo  | US-Wi1  |
| US-ICs  | US-Ro5  | US-SRG  | US-OWC  | US-Tw5  | US-Ton  | US-Hn3  | US-Wi3  |
| US-ICt  | US-Ro6  | US-SRM  | US-Oho  | US-Twt  | US-Var  | US-MMS  | US-Wi5  |
| US-KLS  | US-Rwf  | US-Wjs  | US-Rms  | US-WPT  | US-WCr  | US-PFa  | US-Wi7  |
| US-Mpj  | US-Rws  | US-Wkg  | US-Rwe  | US-Wi6  | US-Wi2  | US-Sta  | US-Wi8  |
| US-Syv  | US-SRC  | ZM-Mon  | US-Whs  | US-xBR  | US-Wi4  |         | US-Wi9  |

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
