# Peer review of "On the added value of sequential deep learning for upscaling of evapotranspiration"

_EGUsphere, 2024_

## Author Response (AR1)

**"On the added value of sequential deep learning for upscaling evapotranspiration"**

**Author's response, 1st iteration, Manuscript #2024-2896**

B. Kraft

May 9, 2025

Dear Editor, dear Reviewers,

Thank you very much for your constructive feedback and valuable suggestions. We are confident that the manuscript has improved significantly as a result of the revisions made in response.

Following the recommendations of the Editor and Reviewers, we included precipitation as an additional covariate. This required rerunning the site-level cross-validation. Precipitation improved certain aspects of the ET predictions, and given our earlier speculations about the influence of PFTs on upscaling, we also conducted a second upscaling run incorporating all covariates (remote sensing, meteorology, precipitation, and PFTs). These results allowed us to further disentangle the sources of uncertainty.

Additionally, based on a Reviewer's suggestion, we performed the site-level cross-validation using the top six models identified during hyperparameter tuning. This provided insights into model robustness.

We acknowledge that these changes go beyond the scope of a minor revision. However, the main message of the manuscript remains unchanged. The results are now more robust, and the discussion has been expanded accordingly.

Please find below our detailed responses to the Reviewers.

Sincerely,
Basil Kraft, on behalf of all co-authors

**Detailed response to Reviewer 1**

> L15: Consider adding quantitative information when discussing model performance differences between the sequential and non-sequential models to make the comparative strengths more transparent for readers.

We updated the abstract, it now contains quantitative information about the model performance differences between the sequential and non-sequential models. The updated part now reads (lines 12-17):

> When using only meteorological covariates, we found that the sequential models (LSTM and TCN) performed better—each with a Nash-Sutcliffe modeling efficiency (NSE) of 0.73—than the instantaneous models (FCN and XGBoost)—both with an NSE of 0.70—in site level cross-validation at the hourly scale. The advantage of the sequential models diminished with the inclusion of remote-sensing-based predictors (NSE of 0.75 to 0.76 versus 0.74). On the anomaly scale, the sequential models consistently outperformed the non-sequential models across covariate setups, with an NSE of 0.36 (LSTM) and 0.38 (TCN) versus 0.33 (FCN) and 0.32 (XGBoost) when using all covariates.

> Lag variables in non-sequential models: Have you considered explicitly adding lagged variables to the non-sequential models? This would offer a more balanced comparison, as it would simulate past dynamics without the complexity of models like LSTM. For example, incorporating lagged climate variables into XGBoost could provide insights into whether sequential models are uniquely beneficial in capturing temporal patterns.

We intentionally did not include lagged variables, as not having to deal with the manual feature crafting is one of the main reasons for using sequential models. Instead, we used proxy variables for system state, such as

vegetation indices. While these cannot entirely directly replace lagged meteorological variables, they offer a simpler experimental setup. Designing lagged features for non-sequential models, as highlighted in studies like Tramontana et al. (2016; https://doi.org/10.5194/bg-13-4291-2016), is complex and presents its own challenges.

> Model selection—self-attention models: The paper mentions TCN and self-attention as alternatives to LSTM, yet only TCN was tested. Could you elaborate on why the self-attention models were excluded from the experiments

Based on our experience, attention-based models typically require more training data than what is available in our case, leading us to expect a lower chance of successful training. There are numerous sequential deep learning models (e.g., gated recurrent units (GRU), attention-based models, convolution-based architectures). We chose LSTM and TCN as representative examples. We explicitly mention this now in the updated manuscript, lines 123-125:

> Two sequential models, one based on the LSTM architecture, and another based on a TCN, account for temporal effects. We chose these models as they are conceptually different but both commonly used for time-series simulation and forecasting tasks, and we acknowledge that other architectures could be used as well.

> L127-129: Precipitation appears to be absent from the meteorological variables. Was there a reason for this omission? Precipitation likely impacts ET indirectly through soil moisture, which could be a significant predictor in capturing temporal dynamics.

We agree that precipitation has a significant effect on ET. Based on your suggestion and the Editor's and other Reviewers' recommendation, we have added precipitation as an additional covariate. We decided to use precipitation from ERA5 reanalysis as additional input and added two new covariate setups: meteorological variable plus precipitation and all covariates (remote sensing, meteorology, precipitation, and PFTs). For the setup with all covariates, we performed a second upscaling run. For more details, please refer to the updated data description (Secion 2), Table 1 showing the covariate setups, and generally the updated results and discussion.

> L129: For clarity, it would help readers if you briefly explained the significance of the time derivative of potential shortwave irradiation and what processes it represents within the context of ET.

We updated the manuscript to clarify this aspect, it now reads (lines 149-151):

> Note that the time-derivative, which is the difference between potential shortwave irradiation values for two consecutive hours, are intended to help the non-sequential models discern the diurnal cycle.

> L175: The statement "The remote sensing and PFT covariates were repeated in time to obtain uniform inputs" could be clarified. Does this mean daily remote sensing data were kept constant at a sub-daily scale? If so, it's worth discussing if the model accounts for the sub-daily variations, as metrics like LST and NDWI are not entirely invariant within a day.

Yes, remote sensing variables were kept constant at a sub-daily scale because these variables do not provide diurnal observations. Diurnal variations are driven primarily by hourly-varying meteorological variables, though they interact with satellite-based features that change only on a daily to weekly basis. We updated the manuscript to clarify this point, and it now reads (lines 203-205):

> Although the remote sensing covariates do, in theory, vary on a sub-daily basis, these variables are not available at the hourly resolution, and the diurnal variations are driven primarily by hourly-varying meteorological variables, though they interact with satellite-based features that change only on a daily to weekly basis.

> L204-205: Were the same hyperparameters used for each fold in cross-validation? This clarification would help assess whether the variation within model ensembles arises from differences in training data subsets or distinct hyperparameter settings.

Yes, hyperparameters were tuned once for each model and setup, and they remained constant throughout cross-validation. Thus, the variability within the model ensemble is attributed to differences in the training data and model optimization, rather than variations in hyperparameter settings. We updated the manuscript to clarify this point, and it now reads (lines 230-232):

> The same hyperparameter set was used throughout the cross-validation for each model setup. Thus, the cross-validation ensemble is composed of models with the same hyperparameters, but trained on different subsets of the data.

*Fig 4—PFT impact on TCN and LSTM models: In Figure 4, adding PFT as a predictor seems to penalize TCN performance, yet it enhances LSTM's accuracy in capturing interannual variability. Could you discuss this divergence?*

Considering the scale of the IAV differences (LSTM improves around 0.05 NSE and TCN decreases around 0.01 NSE), this could just be related to model uncertainty. We updated Figure 4, as you suggested in the next comment, and indeed the small differences are due to uncertainty (see lines 272-278).

*Fig 4—Model sensitivity to hyperparameters: Are the displayed results limited to the best models, or could the performance of the other 19 models (with variation bars) be included to show the sensitivity to hyperparameter choices?*

We updated Figure 4 as you suggested, and included the top six models identified during hyperparameter tuning to assess the model robustness. We did not include all 20 models from hyperparameter tuning as some of these configuration lead to poor performance. We discussed the model robustness in lines 272-278.

*Fig 4—mean-site results: Presenting model performance metrics related to spatial variability, such as the mean site performance, could be informative.*

We have added spatial scale to Figure 4 and discussed the results in lines 269-271 and in a broader context later in the discussion. Adding the spatial scale, and the improved performance when including PFTs, helped us to better understand PFT impact on upscaling.

*L254-255: While PFT doesn't enhance site-level predictions, could it mitigate extrapolation errors during upscaling? Including this consideration in the discussion of the scaling-up section may add valuable insight. Or would the spread for the sequential model change with or without PFT?*

We included an upscaling run with all covariates (remote sensing, meteorology, precipitation, and PFTs) to investigate the influence of PFTs on the upscaling results. Given the complexity of the new findings, we refer the Reviewer to the updated manuscript for a detailed discussion.

*L286-288: Can you test the hypothesis about observation biases with synthetic data or a process-based model simulating extreme events and disturbances? This might strengthen the argument about model vulnerability to changes in predictor distributions.*

We agree with the Reviewer that testing this hypothesis with synthetic data or a process-based model could be valuable. However, this is not a primary focus of the study and would require substantial experimentation.

*L410-411: Jung et al. (2020) introduced an extrapolation index that might be useful here. Plotting the model spread against this index could demonstrate that model uncertainty correlates with areas requiring more extrapolation, supporting your discussion points.*

Unfortunately, applying the method introduced by Jung et al. (2020) is computationally too demanding for our setting (hourly, 0.05degree) due to the need to identify nearest neighbors in the training data for each predicted data point. Adapting this method for large datasets would require significant development and would constitute a separate study. Therefore, we are unable to implement it within the scope of this work.

**Detailed response to Reviewer 2**

*Line 127-129: Could the authors further clarify how meteorological variable combinations are defined? For example, I noticed that precipitation, which often strongly influences ET, is not included as an input feature in their experiments. Could the authors elaborate this?*

The meteorological variables were chosen based on their relevance for ET-related processes. While we agree that precipitation is an important variable for ET and should be included in the analysis, the site-level observations of the variable contain long gaps and filling these gaps is challenging. Therefore, we decided to use precipitation from ERA5 reanalysis as additional input and added two new covariate setups: meteorological variable plus precipitation and all covariates (remote sensing, meteorology, precipitation, and PFTs). For the setup with all covariates, we performed a second upscaling run. For more details, please refer to the updated data description (Secion 2), Table 1 showing the covariate setups, and generally the updated results and discussion.

*Line 129: How did you calculate the time-derivative of potential shortwave irradiation? Please provide more details.*

We updated the manuscript to clarify this aspect, it now reads (lines 149-151):

> Note that the time-derivative, which is the difference between potential shortwave irradiation values for two consecutive hours, are intended to help the non-sequential models discern the diurnal cycle.

Line 136: You mentioned each remote sensing product was interpolated to a daily resolution. Did you use nearest-neighbor interpolation, or another method? Please clarify.

This statement was misleading, and we updated the manuscript to clarify this aspect. The updated part now reads (lines 158-161):

> Although the MCD43A4 product for the reflectances uses observations from a period of 16 days to characterize and invert the bidirectional reflectance distribution function of a given pixel for the day at the center of the period, this operation is done over a temporally moving window at daily timesteps, resulting in output data with daily frequency.

Line 184: The phrase "clustering of coordinates" needs further clarification. It would be helpful to include a table (list of sitenames) summarizing the final results of the 8-fold site clustering in the supplementals.

The clustering was performed randomly, but sites within a 0.05° distance were grouped into the same cluster. We updated the manuscript accordingly (lines 213-215):

> To decrease the dependency between the sets, we ensure that sites in close spatial proximity (below 0.05° distance) are part of the same set using clustering of coordinates.

Following your suggestions, we added Table B1 to the appendix to show the clusters.

Line 191-193: Do you mean that observations from only two random years for each site were selected and used for training, validation, and testing?

We used the full available sequence of observations for training. However, for each training iteration (i.e., each minibatch), a consecutive two-year period was randomly selected from the available data. We updated the manuscript accordingly lines 222-225:

> For a speedup of the training, the model was iteratively fed with randomly selected sequences of two years. The first year was used for providing temporal context similar to the "spinup" in dynamic process models, while the second was used for tuning. Note that the two years were randomly sampled in every epoch, ensuring that all observations were, potentially, used for training with high likelihood.

Line 294-295: If there are discrepancies between EC site observations and the reanalysis dataset, would it be more effective to build a functional relationship between the reanalysis dataset (e.g., extracted ERA5 values at the site level) and ET observations directly, rather than using site-level functional relationships for upscaling? similar to the methodology in Nathaniel (2023). The authors could consider adding a discussion on this point.

Eddy-covariance measurements of land-atmosphere fluxes are unique and rich in information. The processes involved are highly dependent on local conditions, such as land cover (which can be partially captured by remote sensing features) and weather. By relying on ERA5 features, we would lose access to some of this richness, introducing potential biases and missing high-frequency information from the observations. However, using the same data source for both model training and upscaling could reduce biases, which may offer some advantage. While we think that this aspect should be further investigated, we believe that it is out of scope of this study. Also, we concluded in the updated manuscript that the uncertainty of the upscaling is not significantly impacted by the covariate shift due to the different data sources (because where EC sites are present, methods are robust), but rather related to the extrapolation into new and underssamples regions. We refer to lines 390-393 (see excerpt below) and generally to the discussion for a more detailed discussion.

> Because every model and covariate set converges where observational support is strong, the station-to-grid shift cannot be the dominant source of the spatial or global-sum discrepancies. Instead, the residual differences arise from the extrapolation into data-scarse regions, with different behavior across models and covariate setups.

Line 376-378: The sentence, "In summary, the sequential models did, when trained on the same subsets of sites, not behave similar in terms of global annual ET. Therefore, it seems unreasonable to assume that the lower global ET estimated by the sequential neural networks is due to a better (and hence more consistent) representation of the processes," appears ambiguous and redundant. Please revise for clarity and conciseness.

This statement was removed from the manuscript as it was indeed not very clear. A similar statement appears elsewhere in the updated manuscript in lines 378-384, and it reads now:

> Disentangling these possible causes is difficult, but we can draw some conclusions from the results. We have shown that different training subsets did not lead to consistent upscaling behavior (Table 2). In other words, taking the same training data subset for model training did not lead to similar behavior across ML models in terms of global ET. This suggests that the ensemble variance at global scale is not driven by training data, but rather by how the ML models extrapolate out of the training data distribution. It indicates that, when evaluating the global sums alone, neither neural networks versus `xgboost` nor sequential versus non-sequential models can be regarded as inherently more consistent—or therefore more reliable—for global upscaling.